# Decoupled temperature and pressure hydrothermal synthesis of carbon sub-micron spheres from cellulose

Shijie Yu [1,2], Xinyue Dong[3,4], Peng Zhao[1,2], Zhicheng Luo[5], Zhuohua Sun [6], Xiaoxiao Yang[1,2], Qinghai Li[1,2], Lei Wang [3,4✉], Yanguo Zhang [1,2✉] & Hui Zhou [1,2✉]

The temperature and pressure of the hydrothermal process occurring in a batch reactor are typically coupled. Herein, we develop a decoupled temperature and pressure hydrothermal system that can heat the cellulose at a constant pressure, thus lowering the degradation temperature of cellulose significantly and enabling the fast production of carbon sub-micron spheres. Carbon sub-micron spheres can be produced without any isothermal time, much faster compared to the conventional hydrothermal process. High-pressure water can help to cleave the hydrogen bonds in cellulose and facilitate dehydration reactions, thus promoting cellulose carbonization at low temperatures. A life cycle assessment based on a conceptual biorefinery design reveals that this technology leads to a substantial reduction in carbon emissions when hydrochar replacing fuel or used for soil amendment. Overall, the decoupled temperature and pressure hydrothermal treatment in this study provides a promising method to produce sustainable carbon materials from cellulose with a carbon-negative effect.

[1] Key Laboratory for Thermal Science and Power Engineering of Ministry of Education, Department of Energy and Power Engineering, Tsinghua University, Beijing 100084, People's Republic of China. [2] Beijing Key Laboratory of CO2 Utilization and Reduction Technology, Department of Energy and Power Engineering, Tsinghua University, Beijing 100084, People's Republic of China. [3] Key Laboratory of Coastal Environment and Resources of Zhejiang Province, School of Engineering, Westlake University, Hangzhou 310024 Zhejiang, People's Republic of China. [4] Institute of Advanced Technology, Westlake Institute for Advanced Study, Hangzhou 310024 Zhejiang, People's Republic of China. [5] Department of Chemical Engineering and Chemistry, Eindhoven University of Technology, Het Kranenveld 14, Helix, STW 3.48, 5612 AZ Eindhoven, The Netherlands. [6] Beijing Key Laboratory of Lignocellulosic Chemistry, Beijing Forestry University, No.35 Tsinghua East Road, Beijing 100083, People's Republic of China. ✉email: wang_lei@westlake.edu.cn; zhangyg@tsinghua.edu.cn; huizhou@tsinghua.edu.cn

The consumption of fossil fuels continues to produce an increasing amount of $CO_2$ (carbon-positive emission, Fig. 1a), which causes serious consequences such as climate change and ocean acidification. Lignocellulosic biomass, such as wood, grass, and agricultural waste (straw), consisting of cellulose, hemicellulose, and lignin, is a renewable and carbon-neutral resource[1]. The utilization of biomass has great potential in reducing global net carbon emissions[2]. The traditional utilization of biomass, such as combustion, gasification, and anaerobic digestion, is carbon-neutral. The conversion of biomass into carbon materials, which can realize carbon storage in a stable solid form, is a negative emission technology (NET) (Fig. 1a). It has been reported that negative emissions of 7–11 Gt carbon per year are needed in the worst case, and 0.5–3 Gt carbon per year is needed in the best case to meet the 2 °C target[3].

Cellulose, as the main component of lignocellulose biomass (40−60%; mass basis), is also the main component of paper and cotton-based textiles[4]. Cellulose can be converted into carbon materials[5,6], chemicals[7,8], or ethanol[9], whose production are typically highly reliant on fossil fuels. Therefore, the high-value-added utilization of cellulose is expected to contribute to alleviating the energy crisis and global warming. Hydrothermal conversion of cellulose can produce solid carbonaceous materials, liquid bio-oil, and combustible gases (e.g., $H_2$, CO, and $CH_4$)[10–12]. Solid carbonaceous material, i.e., hydrochar, can be used in capacitor electrodes, wastewater treatment, and fuel cells[13,14].

Batch reactors are widely used to study the hydrothermal process of water-insoluble substances due to their easy operation and universality. However, in a typical batch reactor, temperature and pressure are coupled, making it challenging to control them separately, which causes the so-called 'temperature effect' might be essentially a combination of temperature and pressure. Cellulose (crystalline) is generally known to decompose at ~210 °C[15,16] at a saturated vapor pressure of 1.9 MPa. However, when the temperature increases from 100 to 210 °C, the pressure

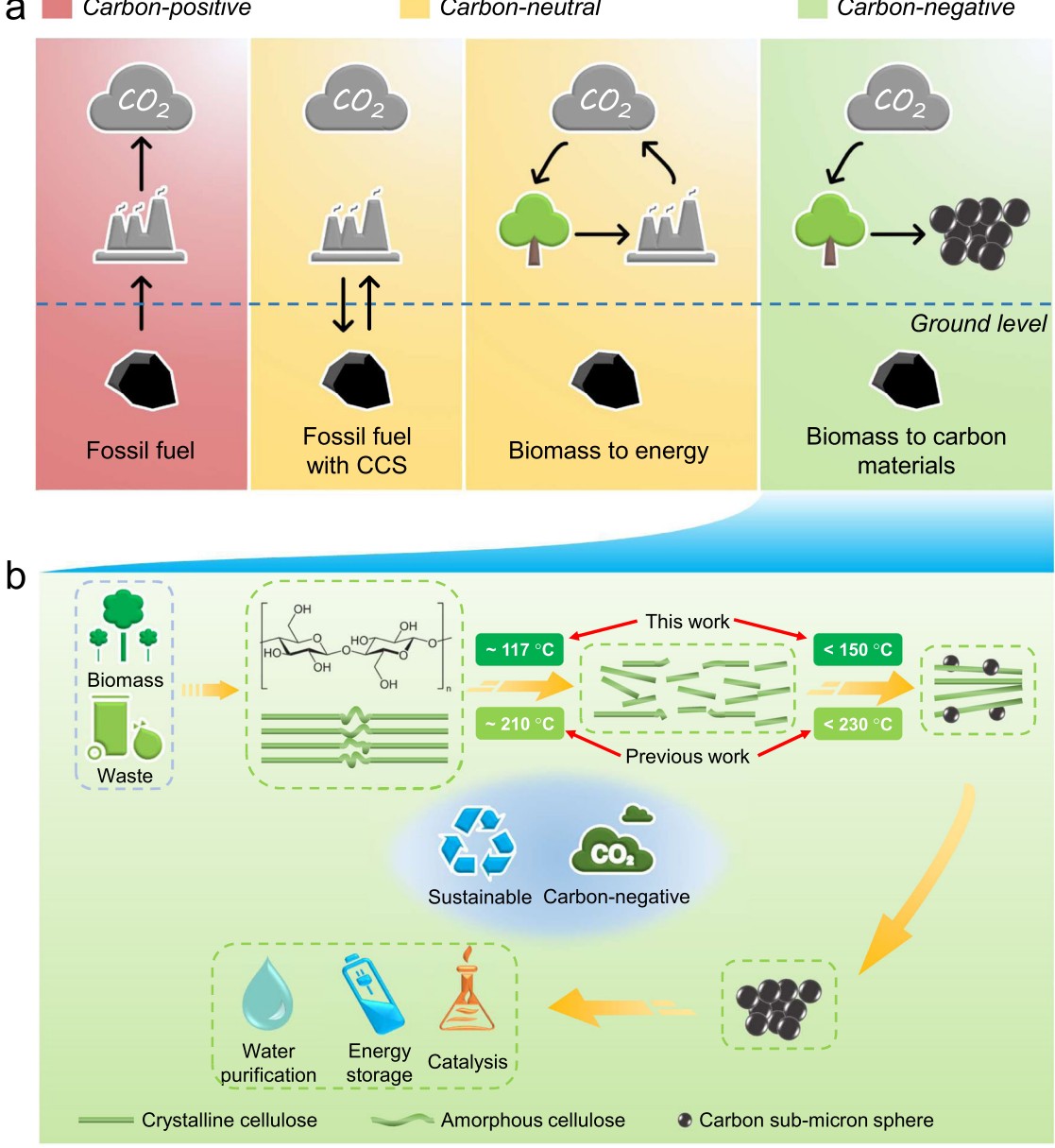

**Fig. 1 Production of carbon sub-micron spheres with a carbon-negative effect. a** Schematic of carbon-positive, carbon-neutral, and carbon-negative situations. **b** Illustration of carbon sub-micron spheres from the low-temperature hydrothermal treatment of cellulose-based feedstocks.

increases from 0.1 to 1.9 MPa, i.e., a coupled temperature and pressure hydrothermal (CTPH) process. Therefore, it is unclear whether this consequence is caused by temperature, pressure, or both. That is, if the pressure changes, the degradation temperature may also change correspondingly.

In this study, we developed a decoupled temperature and pressure hydrothermal (DTPH) system to study the carbonization of cellulose. After separately investigating the temperature (100–300 °C) and pressure (2–20 MPa) in this unique reactor, it was found that the cellulose degrades at ~117 °C and the carbon sub-micron spheres begin to form below 150 °C at a constant pressure of 20 MPa, which was much lower than previous reported temperatures (Fig. 1b; Supplementary Table 1). Carbon sub-micron spheres were produced without any isothermal time, much more quickly as compared to the conventional process that takes several hours[15]. A reaction mechanism was proposed after characterizing the solid products with various technologies. Last, the participation mechanisms of water were proposed through isotopic experiments in $D_2O$. This method shows enhanced energy efficiency compared to conventional methods and a substantial reduction in greenhouse gas emissions based on a prospective life cycle assessment (LCA) of a conceptual biorefinery design. Using this sustainable and carbon-negative technology, we developed a new approach to realizing low-temperature and high-value-added utilization of cellulose and potentially other biomass feedstocks (Fig. 1b).

## Results

**Low-temperature fast production of carbon sub-micron spheres**. Independent control of temperature and pressure can greatly improve the parameter range of the hydrothermal process, i.e., from the saturation line for water to the whole area above this line (Fig. 2a). In this study, a DTPH system was built using a pressure stabilization system to control the pressure (Supplementary Figs. 1 and 2). The mass loss of cellulose was investigated in the DTPH process at 20 MPa (Fig. 2b). Cellulose began to decompose at 117 °C at a mass loss of 3.8%, and a rapid increase in mass loss (from 3.8 to 49.9%) was observed from 117 to 150 °C. This phenomenon revealed that the rapid hydrothermal reaction of cellulose occurred from ca. 117 °C (20 MPa), which was around 100 °C lower than our control experiment of the conventional CTPH process (210 °C) and previously reported results of 210–220 °C (pressure not provided)[15,16]. Such a low reaction temperature was probably enabled by the high reaction pressure, owing to the DTPH system.

To verify this hypothesis, another control experiment at 200 °C and 2 MPa was carried out. Compared with a mass loss of 56.5% at 20 MPa, the mass loss at 2 MPa was only 6.2% (Fig. 2b), suggesting that high pressure significantly promotes the hydrothermal reaction of cellulose. The high carbon content of the solid products from high pressures indicated the carbonization reaction (Supplementary Table 2). According to the results of transmission-based Fourier transform infrared (FTIR) spectroscopy (Fig. 2c), the solid product at 2 MPa retained the initial structure of cellulose well. After hydrothermal treatment at 20 MPa, the OH (1350−1260 and 3700−3000 cm$^{-1}$)[17], C–OH (1100−1000 cm$^{-1}$)[17], and aliphatic C–H (3000−2700 cm$^{-1}$)[18] functional groups in cellulose disappeared with the formation of aromatic C–H (900−720 cm$^{-1}$)[19], C=C (1615 cm$^{-1}$)[20], and C=O (1710 cm$^{-1}$)[17]. The Raman spectra also showed cleavage of the initial skeletal structure (C–C–C, C–O, C–C–O, C–C, C–O–C, and OH; 270−510, 1000−1200, and 3200−3500 cm$^{-1}$)[21,22] and the formation of aromatic structures (1450 and 1600 cm$^{-1}$)[23] in hydrothermally treated cellulose under 20 MPa compared to the original cellulose and

hydrothermally treated cellulose under 2 MPa (Fig. 2d). Cellulose exhibited three peaks at 15.0° (101), 22.5° (002), and 34.5° (040) in X-ray diffraction (XRD) patterns, which are related to the transverse arrangement of microcrystals and the longitudinal structure of the polymer[24]. The crystalline structure still existed under 2 MPa but almost disappeared under 20 MPa (Fig. 2e). Moreover, hydrothermally treated cellulose under 20 MPa showed significantly higher thermal stability than that under 2 MPa in the thermogravimetric analysis (TGA), which was similar to the original cellulose (Fig. 2f).

The above results demonstrate that high pressure significantly promotes the conversion of cellulose (see Supplementary Figs. 3–12 and Supplementary Table 2 for more detailed effects of pressure). From the color of the solid products and the scanning electron microscopy (SEM) images (Fig. 2g–i), carbon sub-micron spheres were well-formed under 20 MPa. The surface area of the solid products also increased from 1.5 to 13.4 m$^2$ g$^{-1}$ with the increased pressure from 2 to 20 MPa (Supplementary Fig. 10). It should be noted that the carbon sub-micron spheres were produced without any isothermal time, much faster compared to the conventional CTPH process, which requires greater than 2 h[15,25]. The fast production of carbon sub-micron spheres makes it more feasible for industrial continuous production. Additionally, the activation energy of the hydrothermal reaction in the DTPH system was 112 kJ mol$^{-1}$ (Supplementary Fig. 15), which is much lower than that of conventional hydrothermal reactions of cellulose (ca. 150 kJ mol$^{-1}$)[26,27], further demonstrating an advantage of the DTPH system.

The hydrothermal process developed in this work was compared with the conventional pyrolysis process (Fig. 2b). Interestingly, the conversion was greatly improved in the hydrothermal process, suggesting that water significantly promoted the decomposition of cellulose. High pressure may amplify the contribution of the aqueous environment to the thermochemical decomposition of cellulose. In contrast to starch, cellulose could not be dissolved in water even under subcritical conditions (300 °C and 25 MPa)[28]. Therefore, the lower decomposition temperature of cellulose in water is not due to the transition from heterogeneous reaction to homogeneous reaction but related to the specific aqueous environment (vide infra). Interestingly, there is no significant difference between the physical properties of water at 200 °C for 2 MPa and 20 MPa (Supplementary Fig. 13), suggesting the facilitation effect of pressure on cellulose conversion comes from other aspects, which will be discussed in later sections. It should be noted the promotion effect of high pressure is also applicable to real biomass feedstock (Supplementary Fig. 14).

**Evolution from cellulose to carbon sub-micron spheres under high pressure**. After the rapid mass loss increase from 3.8 to 49.9% from 117 to 150 °C mentioned above, the mass loss further increased to 62.3% at 250 °C (Fig. 2b). Interestingly, a slight decrease in mass loss from 62.3 to 58.1% was observed from 250 to 300 °C, which was ascribed to the repolymerization of liquid monomers[29]. The product at 100 °C had the appearance of white powder, similar to raw cellulose. In contrast, the products at higher temperatures were brownish-black powders (Supplementary Fig. 16). The surface area of the solid products increased from 7.0 to 27.8 m$^2$ g$^{-1}$ from 100 to 300 °C (Supplementary Fig. 17). The original cellulose showed irregular rod shapes in the SEM imaging, and the destruction of the smooth surface was observed at 100 °C (Supplementary Fig. 18), due to the degradation of amorphous cellulose[30], which was further confirmed by the XRD (vide infra). With the temperature increasing to 150 °C,

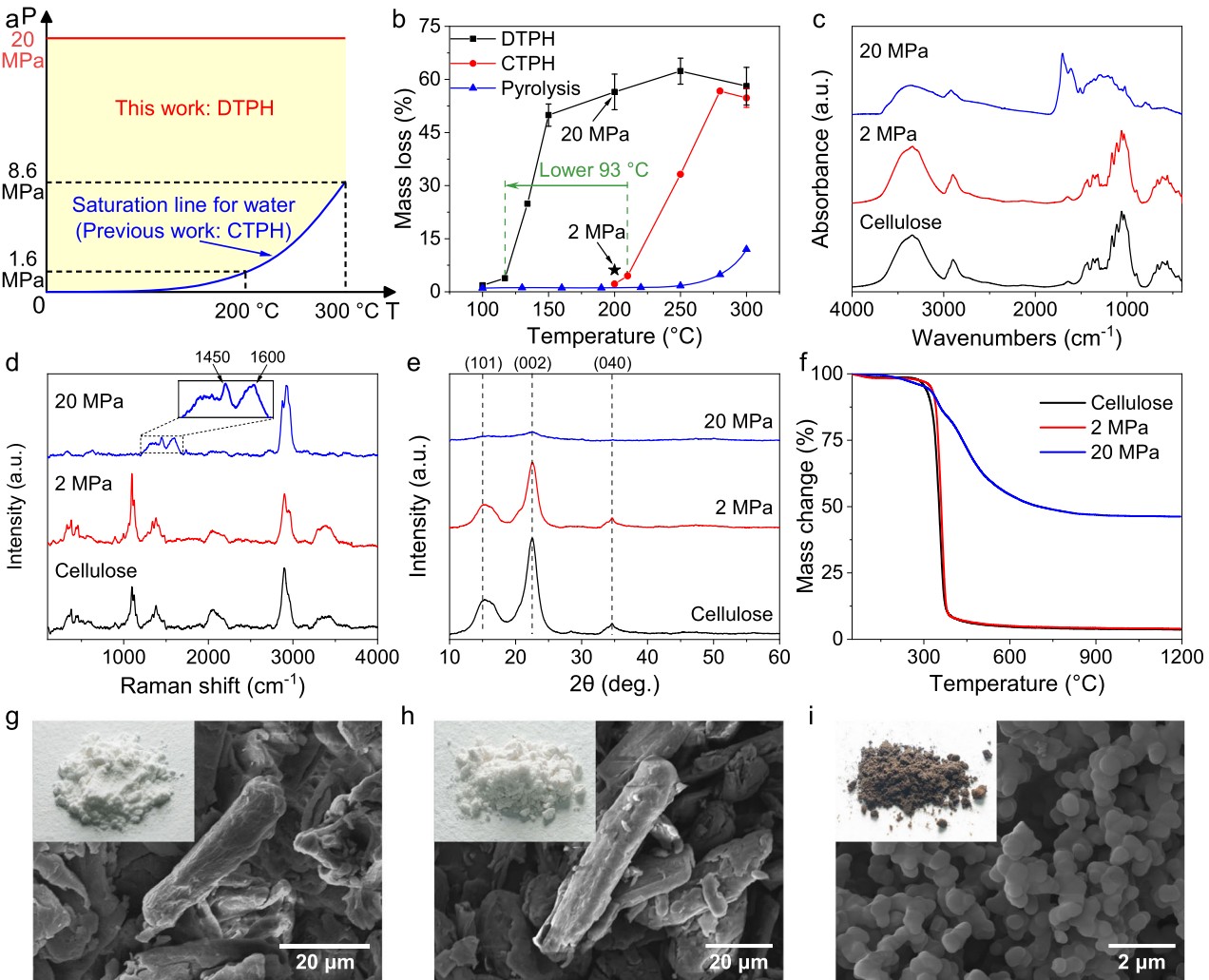

**Fig. 2 Production of carbon sub-micron spheres from cellulose in the DTPH reaction. a** Schematic of obtainable reaction parameters in the DTPH and CTPH process. **b** Mass loss of cellulose in the DTPH process, the CTPH process, and the conventional pyrolysis process. The reaction pressure of the DTPH process was 20 MPa. Pentagram: Mass loss in the DTPH process under 2 MPa. Error bars represent standard deviations of repeated tests. **c**–**f** Comparison of the structures of cellulose, hydrothermally treated cellulose from 200 °C under 2 MPa, and hydrothermally treated cellulose from 200 °C under 20 MPa. **c** FTIR spectra. **d** Raman spectra. **e** XRD patterns. **f** Pyrolysis behavior in TGA. SEM of original cellulose (**g**), hydrothermally treated cellulose from 200 °C under 2 MPa (**h**), and hydrothermally treated cellulose from 200 °C under 20 MPa (**i**). Insets: macroscopic morphologies.

the rod shapes began to degrade, accompanied by the formation of sub-micron spheres on the surface. The rod structures were completely transformed into clusters of spheres with an average diameter of 383 nm at 200 °C (Supplementary Figs. 18 and 19). No obvious change was observed from 200 to 250 °C. Interestingly, the hydrochar generated at 300 °C went through a reshaping process and showed two kinds of spheres. The large spheres had an average size of 531 nm, which may have originated from the coalescence of spheres at 250 °C. The coalescence of carbon spheres increased with temperature from 250 to 300 °C. The small spheres with an average diameter of 109 nm were derived from repolymerization of liquid monomers, confirmed by the carbon spheres from glucose under the same conditions (Supplementary Fig. 20). The sub-micron spheres in this study were much smaller than the microspheres (2–10 μm) generated from the conventional cellulose hydrothermal reaction at 230–250 °C[15], likely due to the DTPH process. Remarkably, similar to the effect of temperature, high pressures could also promote the coalescence of carbon spheres (Supplementary Fig. 9), suggesting that pressure might play a similar role as temperature in the coalescence of carbon spheres.

The carbon content of the hydrothermally treated cellulose increased rapidly from 100 °C (43.0 wt%) to 150 °C (66.7 wt%), and the oxygen and hydrogen content decreased to 28.7 and 4.6%, respectively (Supplementary Table 3). From 150 to 300 °C, the carbon content increased slightly from 66.7 to 76.2 wt%. A Van Krevelen diagram (Fig. 3a) is used to reflect the dehydration, decarboxylation, and demethanation processes[31]. The main process below 100 °C was decarboxylation, and the main process from 100 to 300 °C was dehydration with a small extent of decarboxylation, which was also verified by the results of FTIR, Raman, and XPS (vide infra). In this process, ether, anhydride, and lactone bonds can be formed[31].

To investigate the organic functional groups in the hydrothermally treated cellulose, FTIR analysis was performed (Fig. 3b). The FTIR spectrum of the hydrothermal product at 100 °C was consistent with that of the original cellulose, suggesting that the main structure did not change. In contrast, significant changes were observed from the FTIR spectra of hydrochars obtained at 200 and 300 °C. The bands at 3700−3000 cm$^{-1}$ (free and intermolecularly-bonded hydroxy groups)[32], 1100−1000 cm$^{-1}$ (C−OH stretching)[17], and 1350–1260 cm$^{-1}$ (OH bending)[17]

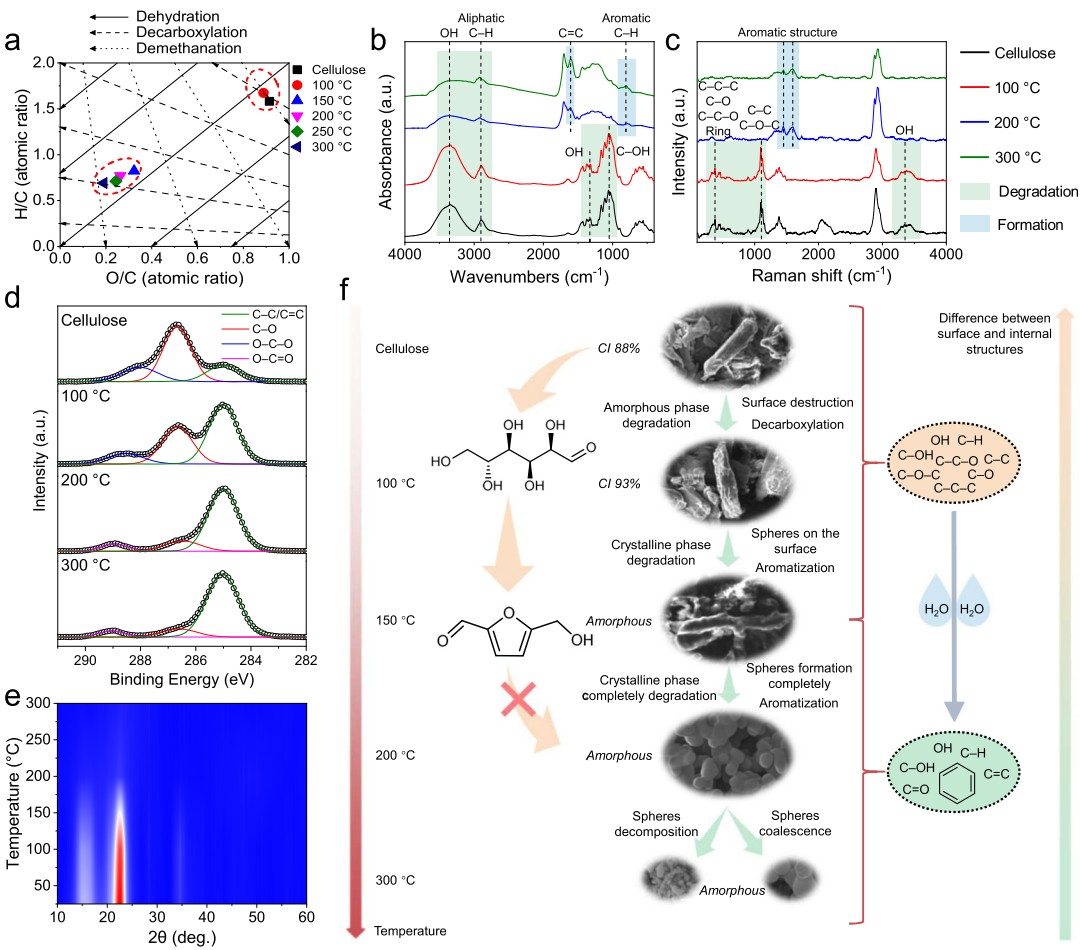

**Fig. 3 Transformation of the cellulose structure in the DTPH reaction. a** Van Krevelen diagram. **b** FTIR spectra. **c** Raman spectra. **d** C 1s XPS. **e** XRD. **f** The mechanism of cellulose transformation in the DTPH process. See Supplementary Figs. 24 and 25 and Supplementary Tables 6 and 7 for the thermal stability.

decreased significantly, indicating dehydration of cellulose in the hydrothermal process. The decrease of the band at $3000-2700\ cm^{-1}$ (aliphatic C–H stretching)[18] and the increases at $900-720\ cm^{-1}$ (aromatic C–H out-of-plane bending)[19] and $1615\ cm^{-1}$ (C=C vibrations)[20] suggest the transformation from an aliphatic structure to an aromatic structure. From 200 to 300 °C, the degrees of aromatization and dehydration were further enhanced.

In the Raman spectra (Fig. 3c), three bands at $270-510\ cm^{-1}$ (C–C–C, C–O, C–C–O, and ring deformation), $1000-1200\ cm^{-1}$ (stretching vibrations of C–C and C–O–C bonds), and $3200-3500\ cm^{-1}$ (OH stretch)[21,22] disappeared in the hydrothermal process from 100 to 200 °C, suggesting fractures of the six-membered rings and the dehydration reaction. Furthermore, two bands at $1450\ cm^{-1}$ (semicircle ring stretch vibration of benzene or condensed benzene rings)[33] and $1600\ cm^{-1}$ (aromatic skeletal vibrations)[23] emerged in the hydrochars of cellulose from 200 °C, which are the typical features of carbonized materials[34].

It has been reported that the surface and internal chemical structures of solid carbon products are different during the hydrothermal treatment of saccharides[35]. To understand the differences between the surface and internal structures, XPS was

performed to investigate the functional groups on the surface (Supplementary Fig. 21). Similar to the elemental analysis results, as the hydrothermal reaction progressed, the surface O/C ratio of the solid significantly decreased (Supplementary Fig. 22). Interestingly, the surface O/C ratio is lower than the overall O/C ratio (surface and internal, from elemental analysis), suggesting cellulose degradation at the surface. It should be noted that the difference in the O/C ratio between the surface and the interior decreased with the extent of reaction, indicating the homogeneous composition of the carbon sub-micron spheres above 200 °C (Supplementary Fig. 22).

In the C 1s spectra, C–C/C=C (285.0 eV), C–O (286.0 eV), O–C–O (288.2 eV), and O–C=O (289.0 eV) were identified (Fig. 3d and Supplementary Table 4)[36,37]. When the hydrothermal temperature reached 100 °C, the fractions of C–O, and O–C–O decreased, while that of C–C increased. Compared with the FTIR spectrum of the solid from 100 °C, which showed negligible change, the significant change of the C 1s XPS indicated surface modification of cellulose via hydrothermal treatment. This result is closely related to the decarboxylation process in the Van Krevelen diagram (Fig. 3a), suggesting that decarboxylation only occurred on the cellulose surface. Surface modification by hydrothermal treatment at 100 °C and 20 MPa, which has not

been reported in the literature, can help to precisely tune the surface properties for heterogeneous catalysis and water treatment. The reaction from 100 to 200 °C formed C=C and C=O bonds, but no significant changes were observed from 200 to 300 °C. The O 1s XPS spectra also confirmed the formation of double bonds from 100 to 200 °C, with the consumption of C−O bonds (Supplementary Fig. 23 and Supplementary Table 5), which is consistent with the FTIR, Raman spectroscopy, and C 1s XPS results.

The XRD pattern of the hydrothermally treated cellulose at 100 °C is similar to that of cellulose (Fig. 3e). The crystallinity index (CI) of original cellulose was 88%, while the CI of the hydrothermal product at 100 °C was 93%, indicating that the amorphous portion of cellulose was more easily decomposed than the crystalline portion at temperatures below 100 °C[38]. The peaks in the XRD patterns decreased significantly when the temperature reached 200 °C, indicating crystallinity loss. A further decrease in crystallinity was observed when the temperature ramped to 300 °C, indicating complete transformation into an amorphous phase.

Based on the above analysis, we proposed a mechanism of cellulose transformation under the DTPH process (Fig. 3f). The flat, rod-shaped cellulose first undergoes a process of amorphous phase decomposition of the surface (decarboxylation) below 100 °C. In the next stage (100−150 °C), the crystalline phase of the cellulose begins to degrade with the formation of spheres on the surface. The residual rod structure is completely converted into uniform spheres with an average diameter of 383 nm at 200 °C. The six-member pyran rings of cellulose are cleaved with the formation of many unsaturated bonds (C=C, C=O, and aromatic C−H). Furthermore, the uniform spheres are reshaped into larger spheres (531 nm) and smaller spheres (109 nm). The formation of spheres is also accompanied by the gradual unification of the surface and internal chemical structures.

**Participation of high-pressure water in the reaction**. To further explore the role of water, we conducted the hydrothermal experiment in $D_2O$ instead of $H_2O$ (200 °C, 20 MPa). As shown by the FTIR (Fig. 4a), compared with the result in $H_2O$, three bands at 3700−3000 cm$^{-1}$ (free and intermolecular bonded OH stretching)[32], 3000−2700 cm$^{-1}$ (aliphatic C−H stretching)[18], and 900−720 cm$^{-1}$ (aromatic C−H out-of-plane bending)[19] decreased in the FTIR spectrum of the material in $D_2O$. Meanwhile, two new peaks, assigned to the OD stretching vibrations (2485 cm$^{-1}$)[39] and C−D stretching vibrations (2335 cm$^{-1}$)[40], can be observed in the results of $D_2O$, suggesting that the O−H and C−H bonds were activated by high-pressure water.

We also performed solid-state $^1H$ nuclear magnetic resonance (NMR) spectroscopy of hydrochars processed at 200 °C and 20 MPa in $H_2O$ and $D_2O$, respectively, to confirm the role of water in the hydrothermal process (Fig. 4b). The peaks were assigned to the hydrogen in the aromatic structure (peak 1), the H in the oxygenated functional groups (−OH, −COOH, −CHO, etc., peak 2), and the H in the aliphatic structure (peaks 3 and 4)[41]. Compared with the hydrochar from $H_2O$, the hydrochar from $D_2O$ showed lower fractions of peaks 1, 2, and 3 and a higher fraction of peak 4. It is known that the absolute intensity of the $^1H$ NMR spectrum of hydrochar in $D_2O$ is lower for the functional groups compared with that in $H_2O$. We could assume a limit case, where the absolute intensity of peak 4 did not change from $H_2O$ to $D_2O$. Taking peak 4 as an internal standard, it was observed that the intensities of peaks 1, 2, and 3 decreased by 49, 72, and 60% from $H_2O$ to $D_2O$ (Fig. 4b and Supplementary Table 8), indicating the substitution of D for H in the aromatic structure, the oxygenated functional groups, and the aliphatic structure, i.e., the participation of water in the hydrothermal conversion of cellulose.

There are two possibilities for the deuterium in $D_2O$ to be present in the hydrochar: the hydrolysis reaction of cellulose or H-D exchange between $D_2O$ and the H in cellulose/intermediates. Most studies have reported that cellulose is hydrolyzed to glucose first, and then glucose is dehydrated to form 5-hydroxymethyl-furfural, which is the core of the formation of carbon spheres[15]. When the same hydrothermal process (200 °C and 20 MPa) of glucose was conducted in this study, no solid products were collected (Fig. 4c). This confirmed that the original cellulose was the direct source of the spheres, and the possibility of direct conversion of cellulose to carbon spheres could be supported by the ref. [29]. It has also been reported that starch was hydrolyzed into monomers and then polymerized into carbon spheres at 180 °C for 24 h[42]. Interestingly, under the conditions of our study (200 °C and 20 MPa), no solid product was collected from the hydrothermal treatment of starch, likely due to the absence of the isothermal time. This result indicates that the formation of carbon sub-micron spheres from cellulose in our study followed a different mechanism compared with the literature, likely due to the lower temperature and higher pressure. In this study, carbon sub-micron spheres were not formed from the condensation of monomers. Indeed, we observed the formation of sub-micron spheres on the surface of the rod cellulose (Supplementary Fig. 18), suggesting that carbon spheres were formed from the original cellulose directly. The formation of the aromatic structure of the hydrochars might be similar to that in the pyrolysis process, i.e., cellulose formed the intermediates through intramolecular and intermolecular rearrangement, which was then converted into an aromatic structure, leading to the formation of char.

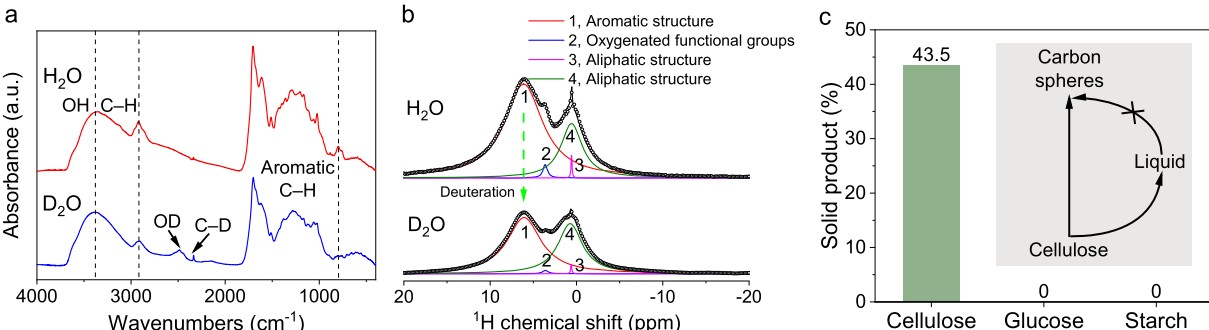

**Fig. 4 Hydrochar characterization from the DTPH reaction in $H_2O$ and $D_2O$. a** FTIR spectra. The stretching vibration band of aromatic C−D is beyond the range of the FTIR (<450 cm$^{-1}$). **b** $^1H$ NMR spectra. Reaction conditions: 200 °C, 20 MPa. **c** Solid product from the hydrothermal process of different substrates at 200 °C and 20 MPa.

Therefore, the D presence in the hydrochar is likely due to the H-D exchange between the $D_2O$ and H in cellulose or the intermediates. The presence of OD in the FTIR and NMR indicates the interactions between the OH in cellulose and high-pressure water. The high-pressure water can help to cleave the inter- and intra-hydrogen bonds and promote the reaction kinetics. Conventionally, the degradation of cellulose needs high temperatures (> 200 °C) to break the refractory hydrogen bonds in the structure[43,44]. In this study, the high-pressure water can help to destroy the hydrogen bonds in cellulose and thus promote the degradation of cellulose at low temperatures (117 °C). The presence of C–D bonds also indicates activation of the C–H bonds in cellulose molecules, leading to the formation of C–C bonds, thus promoting the process of carbonization. It has been reported that the dehydration of cellulose is catalyzed by sulfuric acid[45]. In this study, the high-pressure water may also act as a Brønsted acid catalyst with the release of $H^+$ and $OH^-$ with high energies, which catalyzed the dehydration of cellulose, an essential step in hydrochar formation.

**Sustainability assessment of DTPH carbonization technology.** A conceptual biorefinery plant integrating hydrothermal carbonization using waste cellulose-based feedstock and downstream biogas plant was proposed in this study (Fig. 5a). An energy efficiency assessment based on experimental results and process simulation demonstrates that a higher system energy efficiency could be achieved for the biorefinery design using DTPH technology at 200 °C and 20 MPa compared to that of a conventional CTPH technology at 280 °C and 6.4 MPa (Fig. 5b and Supplementary Fig. 31). As ratios of energy output over input including process energy and those embedded in materials (details are in the Methods section and Supplementary Note 9), the energy efficiencies (EEs) of 62% for wastepaper sludge (WPS, with the main component of cellulose) and 63% for rice straw (RS) also exceed that of biomass pyrolysis reported in other studies (49–51% at 450–650 °C)[46]. LCA was applied for sustainability assessment. The "cradle-to-grave" system boundary starts from the transportation of WPS or the collection of RS till the end use of products as fossil fuel substitution or in soil amendment. The functional unit is the treatment of 1 tonne WPS or RS as received. More details can be found in Supplementary Information for process simulation (Supplementary Note 8), inventory data including inputs/outputs (Supplementary Data 1), and full LCA results (Supplementary Data 1).

Both WPS and RS DTPH carbonization biorefineries can achieve greenhouse gas (GHG) reduction (Fig. 5c) using hydrochar as solid fuel (SF) or for soil amendment (SA). For each tonne of RS being converted to hydrochar, 0.76 tonnes of carbon dioxide equivalent ($CO_2e$) can be reduced for the SF case and 0.30 tonnes $CO_2e$ can be reduced for the SA case. If carbon capture and storage (CCS) is coupled with the DTPH carbonization biorefinery, $CO_2$ from biogas combustion can be captured to further enhance GHG reduction potentials to 1.3 and 0.79 tonnes $CO_2e$ per tonne RS for SF and SA cases, respectively (Supplementary Fig. 29). As residues from one of the major crops, the annual available RS in China is 125 million tonnes distributed mainly in Heilongjiang, Jiangsu, Jiangxi, Hunan, and Hubei provinces (Fig. 5d). RS-derived hydrochar will potentially replace 32.4 million tonnes of coal equivalent (tce) and enable a total of 91.4 million tonnes of $CO_2e$ GHG reduction, positively contributing to China's 2060 carbon neutrality goal.

We also calculated the impacts of the DTPH carbonization biorefinery proposed in this study on other typical environmental indexes (Fig. 5e). Taking hydrochar for solid fuel as an example, the technology is environmentally beneficial in terms of air

quality (ODP), water ecotoxicity (TETP, FETP, and METP), and resource depletion (FFP), for both WPS and RS. It should be noted that the full life cycle processes of the biomass, including the feedstock collection, transportation, and wastewater treatment, are considered in the assessment. Compared to other fossil fuel or biomass-based technologies, the DTPH carbonization in this study shows advantages in both reaction temperature and carbon-negative efficiency as the ratio of carbon from raw materials sequestered in products (Supplementary Fig. 32). Considering the high availability of biomass feedstocks and relatively low energy consumption, DTPH carbonization could be a promising negative emission technology contributing to the global 2 °C target (Supplementary Fig. 33).

## Discussion

In summary, we performed the hydrothermal carbonization of cellulose in a DTPH process. Under a constant pressure of 20 MPa, the surface amorphous structure in cellulose starts to decompose at a temperature lower than 100 °C, and the crystalline structure starts to decompose below 150 °C. Enabled by the DTPH system and the promoting effect of the high-pressure water, carbon sub-micron spheres with smaller diameters were produced in a faster process at lower temperatures compared to previous studies. The role of high-pressure water lies in destroying the hydrogen bonding, activating the C–H bonds, and catalyzing the dehydration, rather than directly participating in the hydrolysis of cellulose. LCAs suggest that this technology enhances energy efficiency and reduces the carbon footprint compared to conventional hydrothermal carbonization pathways. This study may provide new possibilities for the sustainable production of carbon materials and high-value-added utilization of biomass with a carbon-negative effect.

## Methods

**Hydrothermal reaction.** Hydrothermal experiments were carried out in a self-designed DTPH system (Supplementary Fig. 1), including a pressure stabilization system, a connection system, and a reaction system. In the pressure stabilization system, the pump continuously pumped the water from the water tank into the pressure stabilization tank to create stable pressure. The extra water returned to the water tank through a thin tube behind the pressure stabilization tank. The expansion joints and coiling cooler in the connection system could help to block possible material exchange. In the reaction system, a 50 mL Inconel 600 autoclave reactor was heated by an external furnace. The temperature and pressure of the reaction were controlled by the furnace and the pressure stabilization system, respectively, which was the key to the DTPH system.

In the hydrothermal process, 2 g of starting material was loaded in the reactor (see Supplementary Fig. 34 for the influence of loading amount). The reactor was heated from ambient temperature to target temperature under constant pressure (temperature profile shown in Supplementary Fig. 2; see Supplementary Figs. 35 and 36 for the effect of the heating rate). When the temperature reached the target temperature, the reactor was cooled down, and the solid product (hydrothermally treated cellulose) was filtered, washed with abundant water, and dried at 70 °C for 24 h. The experiments were stopped at different temperatures to understand more about each stage of the reaction process. In our typical experiments of 200 °C, the time from room temperature to 200 °C is 21 min, and the time to cool down from 200 to 100 °C is 52 min. The reactor was cooled down from 100 °C to room temperature in an ice water bath in negligible time. The DTPH experiments of rice straw were performed following the same procedure. The CTPH reactions were performed in the same reactor without the pressure stabilization system, and thus the pressure was the self-generated saturated vapor pressure. Mass loss was calculated as:

$$\text{Mass loss} = \left(1 - \frac{M_s}{M_0}\right) \times 100\% \tag{1}$$

where $M_0$ and $M_s$ are the mass of the original cellulose and the solid residue after the reaction, respectively.

**Characterization of the hydrothermally treated cellulose.** Cellulose (microcrystalline, 20 μm), glucose (≥ 99.5%), and starch (from potato) were obtained from Sigma-Aldrich Ltd. Rice straw (from Huaidao #5) was collected from Donghai, Jiangsu Province and ground into powders (< 250 μm). All samples were stored in a desiccator before use. Elemental analyses (C, H, and O) were performed

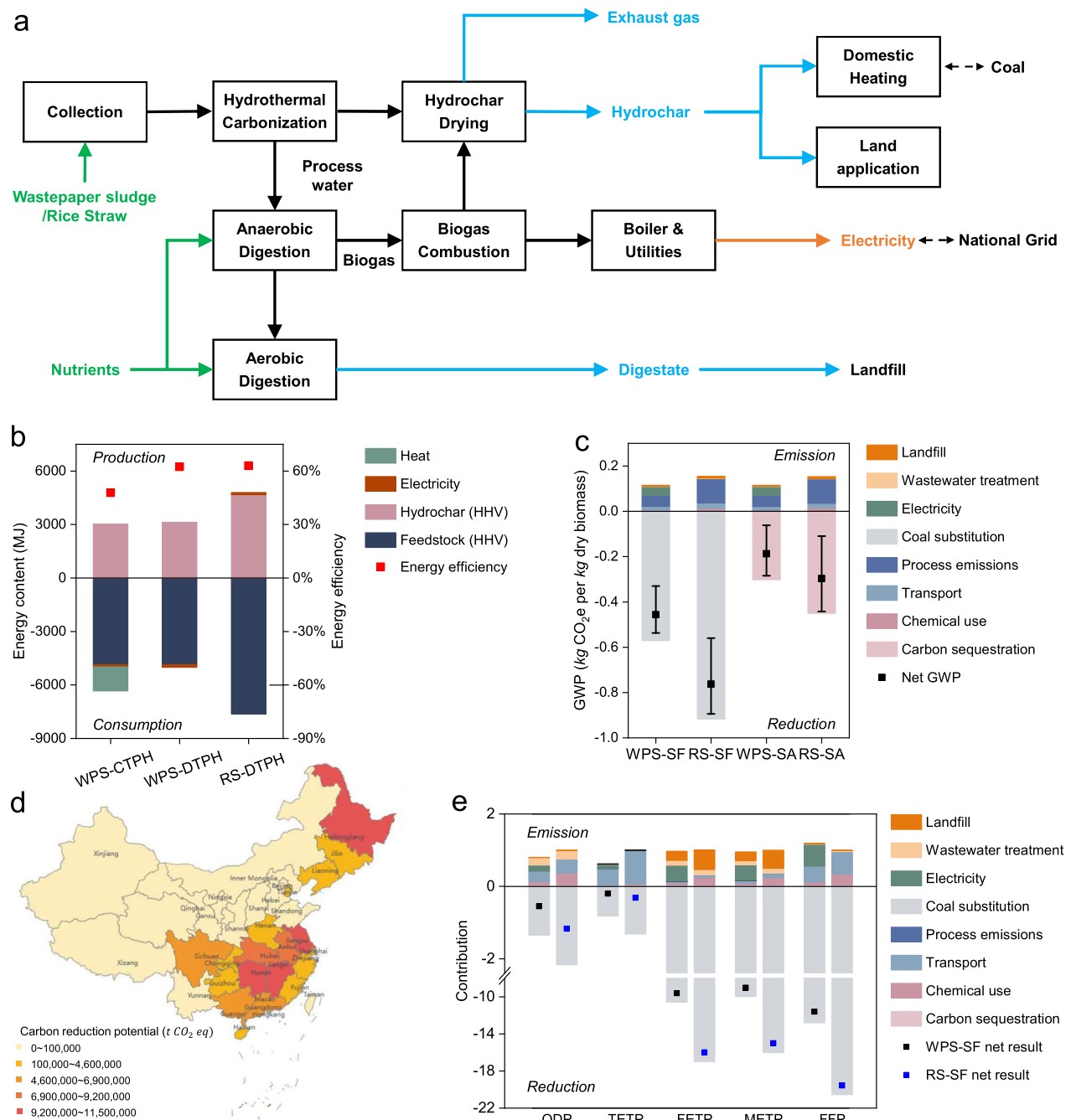

**Fig. 5 Sustainability assessment for wastepaper sludge (WPS) and rice straw (RS) DTPH biorefinery designs. a** System boundary for LCA. **b** Energy efficiency. Reaction condition: CTPH (280 °C, self-generated pressure), DTPH (200 °C, 20 MPa). **c** Global warming potential (GWP) of WPS and RS at 20 wt% biomass/water ratio and 200 °C. SF: solid fuel. SA: soil amendment. Error bars represent the range of net GWP values obtained from various substitution scenarios. **d** GHG reduction potential of RS-SF scenario in China. **e** LCA of WPS-SF and RS-SF at 20 wt% biomass/water ratio and 200 °C normalized based on net values of RS-SF case. ODP: stratospheric ozone depletion (kg CFC-11 eq.). TETP: terrestrial ecotoxicity (kg 1,4-DCB). FETP: freshwater ecotoxicity (kg 1,4-DCB). METP: marine ecotoxicity (kg 1,4-DCB). FFP: fossil resource scarcity (kg oil eq.) See Supplementary Table 11 and Supplementary Data 1 for the full LCA results.

on an Elementar Vario EL III microanalyzer. The transmission FTIR spectra of the samples were recorded on a Nicolet 6700 spectrometer in the range of 4000–400 $cm^{-1}$. The Raman spectra were recorded in the range of 100–4000 $cm^{-1}$ on a Horiba LabRAM HR Evolution spectrometer with a 473 nm excitation laser. The XPS was performed in an ESCALAB 250Xi instrument. The samples were irradiated with Al Kα X-rays (1486.7 eV), and the photoelectrons were analyzed using an HSA type analyzer. The C–C component of C $1s$ (285.0 eV) was used for the calibration. The XRD patterns of the samples were recorded on a Bruker D8 Discover X-ray diffractometer with Cu Kα radiation (40 kV, 40 mA) from 10° to 60°. To compare the crystallinity of the hydrothermally treated cellulose

quantitatively, the crystallinity index (CI) was calculated using the peak height method[47]:

$$CI = \frac{I_{002} - I_{am}}{I_{002}} \times 100\% \qquad (2)$$

where $I_{002}$ is the intensity of the (002) peak and $I_{am}$ is the intensity of the amorphous phase.

The morphologies of the solid materials were investigated with SEM (Zeiss Gemini 300). The surface area of the materials was determined by $N_2$ adsorption/

desorption (Micromeritics ASAP 2460) using the Brunauer-Emmet-Teller (BET) model. The pyrolysis and combustion of the solid materials were performed in a NETZSCH STA 449F3 TGA coupled with FTIR. The samples were heated from room temperature to 1200 °C (30 °C min$^{-1}$) under N$_2$ (100 mL min$^{-1}$) for the pyrolysis experiments and from room temperature to 900 °C (10 °C min$^{-1}$) under air (100 mL min$^{-1}$) for the combustion experiments. The gas products from the TGA experiments were analyzed by a Nicolet Nexus 670 spectrometer (4000–400 cm$^{-1}$, resolution 1 cm$^{-1}$). The solid-state $^1$H NMR spectroscopy was performed on a JEOL JNM-ECZ600R spectrometer (600 MHz $^1$H Larmor frequency), equipped with a 3.2 mm probe operating at room temperature.

**Isotope experiment**. The reaction in D$_2$O (99.9 atom% D, Sigma-Aldrich) was conducted to explore the role of water in the reaction. The participation of water in the reaction could be determined by the substitution of hydrogen by deuterium in the solid products. After the reaction, FTIR and solid-state $^1$H NMR were applied to analyze the substitution of hydrogen by deuterium according to the peak shifts in FTIR spectra and the peak intensity variations in NMR spectra.

**Process simulation**. The whole process includes hydrothermal carbonization, anaerobic digestion (AD), aerobic digestion (AE), biogas combustion, and steam generation (see Supplementary Figs. 26–30 for details). The heat and power demand of the entire process is fulfilled via biogas combustion with supplementation if insufficient. The capacity of the simulation is set as 60,000 tonnes per year, corresponding to a reasonable size of the plant under an economically feasible condition for pyrolysis as a substitution[48]. Scenario processes were simulated in Aspen Plus ®V11 to generate information for the life cycle inventory.

EEs of various scenarios were calculated using the equation below:

$$EE = \frac{E_{hydrochar,HHV} + E_{electricity,out}}{E_{feedstock,HHV} + E_{electricity,in} + E_{thermal,in}} \tag{3}$$

where $E_{hydrochar,\ HHV}$ and $E_{feedstocks,\ HHV}$ are energy contained in the hydrochar and feedstocks. $E_{electricity,\ in}$ and $E_{thermal,\ in}$ represent electricity and heat demand, respectively. In cases where excessive electricity is generated, $E_{electricity,\ out}$ is considered.

**Life cycle assessment (LCA)**. The functional unit is the treatment of 1 tonne of WPS or RS. The process simulations were performed based on a 3000 L h$^{-1}$ reactor, and the designed plant capacity is assumed to be 60,000 tonnes of feedstock per year, reflecting a biorefinery plant with parallel production lines. Mass and energy flows are derived from process simulation (Supplementary Data 1). LCA was conducted in Simapro$^{TM}$ (V9.2). For background data, electricity in China was used in the calculation while relevant chemicals produced in Europe were selected from the Ecoinvent database (v3.0) as substitutions due to the lack of information in China. ReCiPe2016 (H) was applied in the life cycle impact assessment (LCIA) where six environmental impact categories were assessed (Supplementary Data 1). GHG emissions from WPS and RS pathways were quantified via a carbon flow assessment.

## Data availability

The data supporting the findings of this study are available from the corresponding authors upon reasonable request.

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

## Acknowledgements

This research was supported by the National Natural Science Foundation of China (Grant No. 52070116), the Key R&D Program of Guangdong Province (Grant No. 2020B1111380001), the Tsinghua University-Shanxi Clean Energy Research Institute Innovation Project Seed Fund, and the foundation of Westlake University.

## Author contributions

H.Z. conceived the research project. Y.Z. proposed the DTPH concept. S.Y. planned the experimental work. S.Y. and P.Z. built and tested the experimental system. S.Y. prepared, characterized, and tested the materials and analyzed the data. X.D. and L.W. designed and performed the life cycle assessment. L.W. supervised the life cycle assessment. L.W., H.Z., and Y.Z. coordinated the research. Z.L., Z.S., X.Y., and Q.L. provided useful suggestions for the study. Data were discussed among all coauthors. S.Y., H.Z., X.D., and L.W. wrote the paper with contributions from all authors.

## Competing interests

The authors declare no competing interests.
