## [Peer Review File · Nature Communications]

Decoupled temperature and pressure hydrothermal synthesis of carbon sub-micron spheres from celluloseREVIEWER COMMENTS

Reviewer #1 (Remarks to the Author):

Interesting investigation revealing some new insights into hydrothermal carbonization of cellulose by utilizing a reactor where temperature and pressure could be controlled separately and simultaneously. Furthermore authors propose that at higher pressure much lower carbonization temperatures can be utilized, which is expected result but still nicely proven. Authors should address following points:

- 1. The authors state that carbon nanospheres are formed at 150 °C, 20 MPa. Could the authors provide more experimental data and discussion to support this conclusion? Looking at figures, it seems that 200°C and high pressure is needed to achieve carbonization.**
- 2. Amount of starting material used for the processes is missing. This is important to know for the scale up process.**
- 3. "Carbon nanospheres were produced without any isothermal time, much more quickly as compared to the conventional process that takes several hours" With regards to the comment, the authors should state the total time used for the experiments clearly.**
- 4. With regards to all hydrochars produced (independent on temperature and pressure):**
 - a. Are the hydrochars of nanosize or mainly microsize? Authors should consider to rephrasing, for instance to carbon spheres.**
 - b. Could the authors notice any differences in the coalescence of particles depending on pressure and temperature, and the increase/decrease of the parameters? Typically, coalescence of particles increases with temperature.**
- 5. With regards to the constant temperature 200 °C at different pressures: The Raman and XRD in fig 1. are not clear. Is it possible to zoom in over the double peaks (1450 and 1600 cm⁻¹) in Raman? It also seems like there are trace amounts of crystalline cellulose in hydrochars produced at 20 MPa.**
- 6. With regards to the temperature:**
 - a. The authors stated that "A decrease in conversion 163 from 62.3% to 58.1% was observed from 250 to 300 °C, which was ascribed to the repolymerization of liquid monomers²⁹". Could the mass loss be related to further loss of cellulose/oxygen content?**
 - b. At the highest temperature 300 °C the authors showed that two different spheres were produced. Could this be a result of two different mechanisms? Considering that no spheres were formed in case of glucose and starch at lower temperature.**
- 7. About the mechanism: The authors should discuss around the possibility of direct conversion of cellulose to carbon spheres with reference to other works.**
- 8. The procedure of the pyrolysis experiments in experimental section. Is it the same amount of material and reactor that has been used for these experiments?**

Reviewer #2 (Remarks to the Author):

The research idea is simple, however the results are interesting and might contribute to the research in this field. It touches the area that other researchers might ignore. In order to be published, authors have to clarify several issues below:

1. Line 47: Please go over more literatures to confirm the cellulose composition in biomass. The composition ranges should be wider than what authors claimed (42-45%).
2. How author calculated the conversion of cellulose? Please add in the experimental methods
3. Line 149-158: Please strengthen the discussion with the water properties under elevated pressure. What made TPIH much stronger than TPDH as well as pyrolysis? In the manuscript, authors used several common explanations such as "High pressure may amplify the contribution of the aqueous environment to the thermochemical decomposition of cellulose" And "The lower decomposition temperature of cellulose in water is not due to the transition from heterogeneous reaction to homogeneous reaction but related to the specific aqueous environment (vide infra)". However, authors did not explain straight forwardly what made high pressure water can do this work.

In addition, authors should explain the different of physical properties of water at 2 MPa and 20 MPa which might enable the different reaction

4. The reaction mechanism of cellulose conversion under hydrothermal condition has been discussed by several researchers such as Jain, Sevilla. However, they were not discussed specifically the effect of pressure. Since the behavior of TPDH and TPIH as what authors claimed in this research report is different, how about the reaction mechanism? Will elevated pressure lead to different reaction mechanism? Which properties of water dominate the whole reaction mechanism?

The common knowledge in the subcritical water process, the pressure is only maintained at elevated pressure to keep its liquid state.

5. How authors compare OFG in hydrochar synthesized at 200 and 300 oC from FTIR result? Please add the discussion in the manuscript
6. Fig 3b, write the functional group in FTIR graph for easier interpretation such as fig. 4a
7. Please improve the abstract to make it is more critical
8. Authors did experiment without keep the reaction at specific temperature. Then the cellulose conversion occurred at different temperature along with the journey to the specific temperature (Fig 2 supplementary). Then the heating rate will affect the repeatability of this research. Please explain!
And how authors discuss the effect of different temperature during synthesis process?
9. What made TPIH can built nanosphere compare to conventional TPDH? There are several variables may influence, temperature, pressure, time, water state.
10. What is the purpose of hydrochar synthesise from this experiment? Is it used directly as an adsorbent or as an intermediate product for the activated carbon synthesis? Are the hydrochar synthesis from this TPIH meet the requirements for both applications?

Reviewer #3 (Remarks to the Author):

I find very limited novelty in these results. The production of carbon materials from cellulose and also high pressure hydrothermal processes have been reported extensively and the materials thoroughly characterised. See previous work from Sevilla, Titirici, Bacile, etc. LCA is also emerging more and more as a powerful tool to estimate environmental impact. Recently some papers on LCA analysis of hydrothermal carbons

from biomass have been published (*Advanced Energy Materials*, 2200208 and <https://royalsocietypublishing.org/doi/10.1098/rsta.2020.0340>)

I do not find this paper suitable for *Nature Commun*

Reviewer #4 (Remarks to the Author):

This article reports the results of the study of the influence of pressure and temperature on carbonisation of cellulose. In the studied TPIH process cellulose began to decompose at 117 °C which was around 100 °C lower than the control experiment of the conventional TPDH process and carbon nanospheres were produced very quickly as compared to the conventional process. D2O experiments shows that due to the lower temperature and higher-pressure carbon spheres were formed from the original cellulose directly, without the solid phase previously recorded in literature. The article contains also LCA study of conceptual biorefinery plant integrating hydrothermal carbonization using waste cellulose-based feedstock and downstream biogas plant with the goal to assess the environmental impacts of the TPIH carbonization and AD integrated technology systems using wastepaper sludge (WPS) or rice straw (RS) as feedstocks. The conclusions of life cycle assessments suggest that TPIH technology enhances energy efficiency and reduces the carbon footprint compared to conventional hydrothermal carbonization pathways.

The study of the carbonisation of cellulose is complex and the conclusions are supported by enough various data and analyses. The authors clearly explain the novelty of their approach with report to the existing literature. LCA study is also well elaborated and for the improvement of its presentation I recommend the following:

- Short description of the structure of the LCA study should be added in the manuscript.
- Energy efficiency assessment (Fig. 5b) should be clearly explained in the manuscript.
- For different processes should be listed the inputs and outputs in more detail to make analysis more transparent. Especially for the processes of Supplementary Figure 22 which were modelled in Aspen Plus.
- WPS - TPDH scenario should be added in the Fig. 5c and Fig 5e.

Response to comments of Reviewer #1

Interesting investigation revealing some new insights into hydrothermal carbonization of cellulose by utilizing a reactor where temperature and pressure could be controlled separately and simultaneously. Furthermore authors propose that at higher pressure much lower carbonization temperatures can be utilized, which is expected result but still nicely proven. Authors should address following points:

Response

We thank Reviewer 1 for valuable comments on our work and referring to it as “interesting” and “nicely proven”. A point-by-point response to the critical comments of Reviewer 1 is provided below.

Q1. The authors state that carbon nanospheres are formed at 150 °C, 20 MPa. Could the authors provide more experimental data and discussion to support this conclusion? Looking at figures, it seems that 200 °C and high pressure is needed to achieve carbonization.

A1. We have added one more SEM image in Supplementary Figure 18 to show the detailed morphology of the hydrothermally treated cellulose from 150 °C. There are two kinds of morphologies, the formed carbon spheres and the original cellulose skeleton. Carbon spheres begin to form at 150 °C, while cellulose is almost completely converted into carbon spheres at 200 °C. We adjusted this statement in Introduction:

Introduction:

After separately investigating the temperature (100–300 °C) and pressure (2–20 MPa) in this unique reactor, it was found that the cellulose degrades at ~117 °C and the carbon nanospheres **begin to form** below 150 °C at a constant pressure of 20 MPa, which was much lower than previous reported temperatures (Fig. 1b).

Supplementary Figure 18d | SEM of hydrothermally treated cellulose from 150 °C in the TPIH process (20 MPa).

Q2. Amount of starting material used for the processes is missing. This is important to know for the scale up process.

A2. The amount of starting material is 2 g of cellulose, and we also investigated the influence of loading amount, as added in Supplementary Figure 34. We added this information in the manuscript and Supplementary Information. Due to the limitation of the experimental setup,

larger scale experiments cannot be done for the time being. If the device is mature, the experiment can be easily scaled up.

Method:

In the hydrothermal process, 2 g of starting material was loaded in the reactor (see Supplementary Fig. 34 for the influence of loading amount).

Supplementary Figure 34 | Mass loss of cellulose from different amount of starting material.

Q3. “Carbon nanospheres were produced without any isothermal time, much more quickly as compared to the conventional process that takes several hours” With regards to the comment, the authors should state the total time used for the experiments clearly.

A3. In our typical experiments of 200 °C, the time to warm up from room temperature to 200 °C is 21 min (8 °C/min), and the time to cool down from 200 to 100 °C is 52 min. The reactor was cooled down from 100 °C to room temperature in an ice water bath in negligible time. We added this information in Methods.

Q4. With regards to all hydrochars produced (independent on temperature and pressure): Are the hydrochars of nanosize or mainly microsize? Authors should consider to rephrasing, for instance to carbon spheres.

A4. According to our statistical results (Supplementary Figures 8 and 19), the maximum diameter of the carbon spheres does not exceed 1 µm at various reaction conditions, and the average diameters are all less than 600 nm. Therefore, we think it can be defined as carbon nanospheres.

Q5. Could the authors notice any differences in the coalescence of particles depending on pressure and temperature, and the increase/decrease of the parameters? Typically, coalescence of particles increases with temperature.

A5. Thanks for this valuable question. We investigated the influence of temperature and pressure on the coalescence of the carbon spheres, respectively. We added the SEM imaging of the carbon spheres at 250 °C and observed the coalescence of carbon nanospheres increased with temperature (mainly from 250 to 300 °C) (Supplementary Figure 18, shown below). Besides, we added the experiment at 9 MPa (300 °C) and found that the pressure could also promote the coalescence of carbon nanospheres from 9 to 20 MPa (Supplementary Figure 9, shown below). This phenomenon suggested that pressure played a similar role as temperature in the coalescence of carbon nanospheres. The corresponding data and discussion have been added in the revised manuscript and Supplementary Information.

Supplementary Figure 18 | SEM of hydrothermally treated cellulose in the TPIH process (20 MPa). (e) hydrothermally treated cellulose from 200 °C; (f) hydrothermally treated cellulose from 250 °C; (g) hydrothermally treated cellulose from 300 °C.

Supplementary Figure 9 | SEM of hydrochar under different pressures at 300 °C. (a) 9 MPa; (b) 20 MPa.

Q6. With regards to the constant temperature 200 °C at different pressures: The Raman and XRD in fig 2. are not clear. Is it possible to zoom in over the double peaks (1450 and 1600 cm^{-1}) in Raman? It also seems like there are trace amounts of crystalline cellulose in hydrochars produced at 20 MPa.

A6. We zoomed in over the double peaks in Raman and added the peak information both in Raman and XRD:

Fig. 2d. Raman spectra of cellulose, hydrothermally treated cellulose from 200 °C under 2 MPa, and hydrothermally treated cellulose from 200 °C under 20 MPa.

Fig. 2e. XRD patterns of cellulose, hydrothermally treated cellulose from 200 °C under 2 MPa, and hydrothermally treated cellulose from 200 °C under 20 MPa.

We agreed that there are trace amounts of crystalline cellulose in hydrochars produced at 20 MPa. We adjusted the statement about it:

The crystalline structure still existed under 2 MPa but almost disappeared under 20 MPa (Fig. 2e).

Q7. With regards to the temperature: The authors stated that “A decrease in conversion from 62.3% to 58.1% was observed from 250 to 300 °C, which was ascribed to the repolymerization of liquid monomers²⁹”. Could the mass loss be related to further loss of cellulose/oxygen content?

A7. The decrease in conversion from 62.3% to 58.1% means the decrease of mass loss of cellulose or the increase of the yield of carbon nanospheres. The conversion (mass loss) increased with temperature from 100 to 250 °C as expected, but decreased from 250 to 300 °C. We inferred that the opposite trend was due to the repolymerization of liquid monomers, and some similar results were found in the cited reference. This is closely related to the reviewer's next question about the mechanism of formation. We further address it in the next answer. Moreover, to make it clearer, we modified "conversion" to "mass loss" in the text.

Q8. At the highest temperature 300 °C the authors showed that two different spheres were produced. Could this be a result of two different mechanisms? Considering that no spheres were formed in case of glucose and starch at lower temperature.

A8. We agree with the reviewer that the two different spheres at 300 °C are the result of two different mechanisms. As indicated by the hypothesis in the last answer, the opposite trend (decrease) of mass loss at 300 °C was due to the repolymerization of liquid monomers. Therefore, we believe that the smaller spheres at 300 °C were formed by the repolymerization of liquid monomers, and the larger spheres at 300 °C were originated from the coalescence of carbon nanospheres at lower temperatures, which has been confirmed in the previous answer. To confirm this, we added a TPIH experiment of glucose at 300 °C, and a significant amount (28.8%) of carbon nanospheres were observed, and the size of carbon nanospheres was similar to the smaller spheres from cellulose (Supplementary Figure 20, shown below). Therefore, the two different spheres from 300 °C were from two different mechanisms. We added the data and discussion in the manuscript and the Supplementary Information.

Supplementary Figure 20 | SEM of hydrochar from glucose at 300 °C and 20 MPa.

Q9. About the mechanism: The authors should discuss around the possibility of direct conversion of cellulose to carbon spheres with reference to other works.

A9. Thanks for the suggestion. We have added the related reference and discussed in the revised text:

This confirmed that the original cellulose was the direct source of the nanospheres, and the possibility of direct conversion of cellulose to carbon spheres could be supported by the reference²⁹.

Q10. The procedure of the pyrolysis experiments in experimental section. Is it the same amount of material and reactor that has been used for these experiments?

A10. The pyrolysis experiments were performed in the thermogravimetric analyzer (TGA) with the cellulose of 5 mg. At the typical temperature of 200 °C, no cellulose was converted during

the pyrolysis experiment. We added a pyrolysis experiment in our hydrothermal reactor, and the result was the same.

Response to comments of Reviewer #2

The research idea is simple, however the results are interesting and might contribute to the research in this field. It touches the area that other researchers might ignore. In order to be published, authors have to clarify several issues below:

Response

We thank Reviewer 2 for recognizing our results as “interesting and might contribute to the research in this field” and referring to it as “touches the area that other researchers might ignore”. We answer, point-by-point, the critical questions of Reviewer 2 below.

Q11. Line 47: Please go over more literatures to confirm the cellulose composition in biomass. The composition ranges should be wider than what authors claimed (42-45%).

A11. Thanks for pointing this out. We have modified the composition range of cellulose in biomass as follows:

Cellulose, as the main component of lignocellulose biomass (40–60%; mass basis), is also the main component of paper and cotton-based textiles⁴.

Q12. How author calculated the conversion of cellulose? Please add in the experimental methods

A12. We modified “conversion” to “mass loss” and added the definition in the experimental methods:

Mass loss was calculated as:

$$\text{Mass loss} = \left(1 - \frac{M_s}{M_0}\right) \times 100\% \quad (1)$$

where M_0 and M_s are the mass of the original cellulose and the solid residue after the reaction, respectively.

Q13. Line 149-158: Please strengthen the discussion with the water properties under elevated pressure. What made TPIH much stronger than TPDH as well as pyrolysis? In the manuscript, authors used several common explanations such as “High pressure may amplify the contribution of the aqueous environment to the thermochemical decomposition of cellulose” And “The lower decomposition temperature of cellulose in water is not due to the transition from heterogeneous reaction to homogeneous reaction but related to the specific aqueous environment (vide infra)”. However, authors did not explain straight forwardly what made high pressure water can do this work. In addition, authors should explain the different of physical properties of water at 2 MPa and 20 MPa which might enable the different reaction.

A13. We added the physical properties of water at 2 MPa and 20 MPa (Supplementary Figure 13) and the corresponding discussion in the text. There is no significant difference between the physical properties of water at 200 °C for 2 MPa and 20 MPa. However, the increase of the water pressure can promote the reaction kinetics significantly, since water can participate in the reaction.

This suggests that the significant facilitation effect of pressure on cellulose conversion is due to other reasons. We further discussed the mechanism of high-pressure water effect as follows:

“The presence of OD in the FTIR and NMR indicates the interactions between the OH in cellulose and high-pressure water. The high-pressure water can help to cleave the inter- and intra-hydrogen bonds and promote the reaction kinetics. Conventionally, the degradation of cellulose needs high temperatures (> 200 °C) to break the refractory hydrogen bonds in the structure^{43, 44}. In this study, the high-pressure water can help to destroy the hydrogen bonds in

cellulose and thus promote the degradation of cellulose at low temperatures (117 °C). The presence of C–D bonds also indicates activation of the C–H bonds in cellulose molecules, leading to the formation of C–C bonds, thus promoting the process of carbonization. It has been reported that the dehydration of cellulose is catalyzed by sulfuric acid⁴⁵. In this study, the high-pressure water may also act as a Brønsted acid catalyst with the release of H⁺ and OH⁻ with high energies, which catalyzed the dehydration of cellulose, an essential step in hydrochar formation.”

To highlight this discussion, we changed the section title “Participation of water in the reaction” to “Participation of high-pressure water in the reaction”.

Supplementary Figure 13 | Physical properties of water under different pressures (200 °C). (a) Density; (b) thermal conductivity; (c) viscosity.

Interestingly, there is no significant difference between the physical properties of water at 200 °C for 2 MPa and 20 MPa (Supplementary Fig. 13), suggesting the facilitation effect of pressure on cellulose conversion comes from other aspects, which will be discussed in later sections.

Q14. The reaction mechanism of cellulose conversion under hydrothermal condition has been discussed by several researchers such as Jain, Sevilla. However, they were not discussed specifically the effect of pressure. Since the behavior of TPDH and TPIH as what authors claimed in this research report is different, how about the reaction mechanism? Will elevated pressure lead to different reaction mechanism? Which properties of water dominate the whole reaction mechanism? The common knowledge in the subcritical water process, the pressure is only maintained at elevated pressure to keep its liquid state.

A14. As we shown in A13, the properties of water were not significantly different at 2 MPa and 20 MPa. As indicated by the hydrochar FTIR and NMR results from in H₂O and D₂O, the high-pressure water can help to cleave the inter- and intra-hydrogen bonds and promote the reaction kinetics, which is different from the common knowledge that the only role of pressure is to maintain its liquid state. The detailed mechanisms are discussed in A13.

Q15. How authors compare OFG in hydrochar synthesized at 200 and 300 °C from FTIR result? Please add the discussion in the manuscript

A15. The comparison of OFG in hydrochar synthesized at 200 and 300 °C from FTIR result has been added as follows:

From 200 to 300 °C, the degrees of aromatization and dehydration were further enhanced.

Q16. Fig 3b, write the functional group in FTIR graph for easier interpretation such as fig. 4a

A16. Thanks for the suggestion. We have added the functional groups in Fig. 3b. Moreover, we added the structure information in Fig. 3c.

Fig. 3 | Transformation of the cellulose structure in the temperature-pressure independent hydrothermal (TPIH) reaction. b, FTIR spectra. c, Raman spectra.

Q17. Please improve the abstract to make it is more critical

A17. Thanks for the suggestion. We improved the abstract as follows:

Synthesizing carbon materials is limited by the high temperature and elevated carbon emissions. Herein, we developed a temperature-pressure independent hydrothermal (TPIH) system that can heat the cellulose at a constant pressure, thus lowering the degradation temperature of cellulose significantly and enabling the fast production of carbon nanospheres. Carbon nanospheres could be produced without any isothermal time, much faster compared to the conventional hydrothermal process. High-pressure water could help to cleave the hydrogen bonds in cellulose and facilitate dehydration reactions, thus promoting cellulose carbonization at low temperatures. A life cycle assessment based on a conceptual biorefinery design revealed that this technology led to a substantial reduction in carbon emissions when hydrochar replacing fuel or used for soil amendment. Overall, the TPIH treatment in this study provides a promising method to produce sustainable carbon materials from cellulose with a net carbon-negative effect.

Q18. Authors did experiment without keep the reaction at specific temperature. Then the cellulose conversion occurred at different temperature along with the journey to the specific temperature (Fig 2 supplementary). Then the heating rate will affect the repeatability of this research. Please explain! And how authors discuss the effect of different temperature during synthesis process?

A18. The temperature marked in this paper is the final temperature in the TPIH reaction. The hydrothermal process in this study is a dynamic process similar to TGA, without the isothermal process, thus realizing the fast synthesis of carbon spheres. We agree with the reviewer that the heating rate will affect the repeatability of this research. We added the experiments at different heating rates. The results showed that the heating rate had little effect on the yields of carbon nanospheres (Supplementary Figure 35), but affected the particle size of carbon nanospheres. A higher heating rate was more favorable for the production of uniform carbon nanospheres. A lower heating rate led to the formation of smaller carbon nanospheres, which might be attributed to the repolymerization of liquid monomers (Supplementary Figure 36).

Supplementary Figure 35 | Mass loss of cellulose under different heating rates.

Supplementary Figure 36 | SEM of hydrochar under different heating rates.

Q19. What made TPIH can built nanosphere compare to conventional TPDH? There are several variables may influence, temperature, pressure, time, water state.

A19. It is the high pressure that made the TPIH process can build nanospheres compared to the conventional TPDH process. We conducted rigorous control experiments to compare TPIH and TPDH reactions. All experimental conditions for the TPDH process were the same as the TPIH process, except that the pressure stabilization system was not used. Furthermore, we carried out the TPIH experiment at different pressures (2, 4, 6, 8, 14, and 20 MPa) to verify the

promoting effect of high pressure (see Supplementary Note 3). Therefore, we think that the high pressure is the key to produce the nanospheres.

Q20. What is the purpose of hydrochar synthesized from this experiment? Is it used directly as an adsorbent or as an intermediate product for the activated carbon synthesis? Are the hydrochar synthesis from this TPIH meet the requirements for both applications?

A20. The core of this study is to report that high pressure can significantly reduce the hydrothermal carbonization temperature of cellulose, which is a fundamental work and does not involve the application of the carbon materials for the time being. The hydrochar we obtained is as well generalized as previous studies, and the same can be modified and functionalized accordingly for different applications. We think it is promising to apply the hydrochar for different requirements in our subsequent studies.

Response to comments of Reviewer #3

I find very limited novelty in these results. The production of carbon materials from cellulose and also high pressure hydrothermal processes have been reported extensively and the materials thoroughly characterised. See previous work from Sevilla, Titirici, Bacile, etc. LCA is also emerging more and more as a powerful tool to estimate environmental impact. Recently some papers on LCA analysis of hydrothermal carbons from biomass have been published (Advanced Energy Materials, 2200208 and <https://royalsocietypublishing.org/doi/10.1098/rsta.2020.0340>)

I do not find this paper suitable for Nature Commun.

Response

We thank Reviewer 3 for the comments. We summarized the references on the hydrothermal synthesis of carbon spheres from cellulose, including those mentioned by Reviewer 3. It can be seen that the previously reported temperatures for cellulose decomposition are all around 200 °C. Remarkably, in our TPIH process of 20 MPa, the cellulose decomposition temperature is only 117 °C, which is much lower than the conventional TPDH process. The formation temperature of carbon spheres is also reduced. Furthermore, based on the hydrothermal carbonization mechanism of cellulose, we proposed a fast production strategy, which could reduce the long reaction time of hydrothermal carbonization significantly. Moreover, the size of the carbon spheres is much smaller than those in the literature due to the unique TPIH system. We also performed the isotope study to reveal the involvement mechanism of high-pressure water. In conclusion, the novelty of this paper lies in a low-temperature and rapid production process of carbon nanospheres with significantly lower temperature and shorter time compared to previously reported results. We added the comparison in Supplementary Table 1:

Supplementary Table 1 | Comparison between the TPIH reaction of cellulose in this study and the TPDH reactions in the literature.

	Crystallinity index (%)	Initial decomposition temperature (°C)	Initial formation temperature of carbon spheres (°C)	Complete formation temperature of carbon spheres (°C)	Pressure (MPa)	Isothermal time (h)	Yield of carbon spheres (%)	Ref.
TPDH	88 ^a	210	220	230	Not given	4.0	33.5	1
TPDH	81 ^a	180	200	220	Not given	24.0	~37.0	2
TPDH	Not given	200	Not given	250	2 ^b	~2.2	~32.0	3
TPDH	Not given	220	Not given	Not given	Not given	0.5	-	4

TPDH	Not given	Not given	220	220	Self-generated pressure	4.0	46.1	5
TPDH	88	210	270 ^c	280 ^c	Self-generated pressure	0 ^d	43.5	Control experiment
TPIH	88	117	150	200	20 ^e	0 ^d	43.5	This work

^a Calculated by the XRD pattern in the ref.; ^b initial pressure before heating in the TPDH process; ^c obtained based on the same mass loss as the TPIH process; ^d there is no isothermal time; the total time for heating and cooling was 1.2 h; ^e constant pressure in the TPIH process.

References

1. Sevilla M, Fuertes AB. The production of carbon materials by hydrothermal carbonization of cellulose. *Carbon* 47, 2281-2289 (2009).
2. Falco C, Baccile N, Titirici MM. Morphological and structural differences between glucose, cellulose and lignocellulosic biomass derived hydrothermal carbons. *Green Chem.* 13, 3273-3281 (2011).
3. Gao Y, Wang XH, Yang HP, Chen HP. Characterization of products from hydrothermal treatments of cellulose. *Energy* 42, 457-465 (2012).
4. Kim D, Lee K, Park KY. Upgrading the characteristics of biochar from cellulose, lignin, and xylan for solid biofuel production from biomass by hydrothermal carbonization. *J. Ind. Eng. Chem.* 42, 95-100 (2016).
5. Sheng KC, et al. Hydrothermal carbonization of cellulose and xylan into hydrochars and application on glucose isomerization. *J. Cleaner Prod.* 237, 117831 (2019).

In addition to technology development, our work incorporates a prospective LCA using inventory generated via Aspen Plus to support its sustainability, strengthening the comprehensiveness. Furthermore, we integrate HTC with a downstream AD process to complete the entire design rather than focusing on the carbonization process itself, enabling energy and material optimization towards more realistic circumstances.

Response to comments of Reviewer #4

This article reports the results of the study of the influence of pressure and temperature on carbonisation of cellulose. In the studied TPIH process cellulose began to decompose at 117 °C which was around 100 °C lower than the control experiment of the conventional TPDH process and carbon nanospheres were produced very quickly as compared to the conventional process. D2O experiments shows that due to the lower temperature and higher-pressure carbon spheres were formed from the original cellulose directly, without the solid phase previously recorded in literature.

The article contains also LCA study of conceptual biorefinery plant integrating hydrothermal carbonization using waste cellulose-based feedstock and downstream biogas plant with the goal to assess the environmental impacts of the TPIH carbonization and AD integrated technology systems using wastepaper sludge (WPS) or rice straw (RS) as feedstocks. The conclusions of life cycle assessments suggest that TPIH technology enhances energy efficiency and reduces the carbon footprint compared to conventional hydrothermal carbonization pathways.

The study of the carbonisation of cellulose is complex and the conclusions are supported by enough various data and analyses. The authors clearly explain the novelty of their approach with report to the existing literature. LCA study is also well elaborated and for the improvement of its presentation I recommend the following:

Response

We thank Reviewer 4 for the positive evaluation of our work, recognizing that “the conclusions are supported by enough various data and analyses”, “the authors clearly explain the novelty of their approach with report to the existing literature”, and “LCA study is also well elaborated”. A point-by-point response to the comments of Reviewer 4 is provided below.

Q21. Short description of the structure of the LCA study should be added in the manuscript.

A21. The description of the LCA structure was illustrated in Fig.5a as a system boundary definition. It describes the starting point of LCA as the collection of straw till the endpoint as the use of hydrochar as a solid fuel or soil amender. More details, including goal and scope definition, life cycle inventory, and life cycle impact assessment, have been provided in Supplementary Note 8. In order to facilitate the ease of reading, we added a brief description in the main text.

LCA was applied for sustainability assessment. The “cradle-to-grave” system boundary starts from the transportation of WPS or the collection of RS till the end use of products as fossil fuel substitution or in soil amendment. The functional unit is the treatment of 1 tonne WPS or RS as received. More details can be found in Supplementary Information for process simulation (Supplementary Note 8), inventory data including inputs/outputs (Supplementary Excel Sheet D), and full LCA results (Supplementary Excel Sheet E&F).

Q22. Energy efficiency assessment (Fig. 5b) should be clearly explained in the manuscript.

A22. The assessment of energy efficiency (EE) was described in the Methods section. The EE is the ratio as the sum of energy embedded in products and surplus energy output, if any, over the energy amount of raw materials plus energy input, as stated in equation (3). More details about the method and full EE assessment results have been given in Supplementary Note 9. Again, we added more text to explain the method briefly and point to SI.

As ratios of energy output over input including process energy and those embedded in materials (details are in the Methods section and Supplementary Note 9), the energy efficiencies of 62% for wastepaper sludge (WPS, with the main component of cellulose) and 63% for rice straw (RS) also exceed that of biomass pyrolysis reported in other studies (49%–51% at 450–650 °C)⁴⁶.

Q23. For different processes should be listed the inputs and outputs in more detail to make analysis more transparent. Especially for the processes of Supplementary Figure 22 which were modelled in Aspen Plus.

A23. Due to the limited space in the main text, we have listed the inputs and outputs of Aspen plus in the Supplementary Excel Sheet (Sheet A for mass and Sheet B for energy). Life cycle inventory is listed in Sheet D. We have added in the main text to point it out (the same as A21).

Q24. WPS - TPDH scenario should be added in the Fig. 5c and Fig 5e.

A24. We appreciate this valuable suggestion. With regards to LCA results, energy is normally one of the main contributors to the environmental burdens, as revealed in numerous existing studies as well as in our WPS-TPIH cases (positive scores in Fig 5c and 5e) where electricity is required as input. As shown in Fig.5b, due to lower energy efficiency for WPS-TPDH, the process is not energy self-sufficient and requires extra heat. Therefore, as expected, environmental burdens for WPS-TPDH shall be higher than WPS-TPIH case. For LCA results to be concise, we prefer to structure them as a comparison between different use of final products under the same reaction conditions.

REVIEWERS' COMMENTS

Reviewer #1 (Remarks to the Author):

Authors have clearly improved the manuscript. I still have one issue concerning the naming:

Q4/A4 in response-to-reviewers. The definition of nanosized is typically 1-100 nanometer and at least one dimension should be under 100 nm, so in my opinion these are carbon spheres and not carbon nanospheres.

Reviewer #2 (Remarks to the Author):

The authors has responded to the reviewer comments. The finding in this paper need to be countered by other work. A further discussion with other papers that refer to this paper will be interesting.

Reviewer #4 (Remarks to the Author):

The authors addressed all my recommendations. This makes the text of manuscript clearer. As for the electricity consumed (excel sheet B), only the aggregate value is given for each weight ratio.

Response to comments of Reviewer #1

Authors have clearly improved the manuscript. I still have one issue concerning the naming:

Q4/A4 in response-to-reviewers. The definition of nanosized is typically 1-100 nanometer and at least one dimension should be under 100 nm, so in my opinion these are carbon spheres and not carbon nanospheres.

We thank Reviewer #1 for the positive evaluation of our work. We have modified all the related statements of carbon nanospheres into sub-micron spheres.

Response to comments of Reviewer #2

The authors has responded to the reviewer comments. The finding in this paper need to be countered by other work. A further discussion with other papers that refer to this paper will be interesting.

We thank Reviewer #2 for the valuable comments on our work. We are also looking forward to discussing with future related studies.

Response to comments of Reviewer #4

The authors addressed all my recommendations. This makes the text of manuscript clearer. As for the electricity consumed (excel sheet B), only the aggregate value is given for each weight ratio.

Thanks for pointing it out. The process simulation for each weight ratio case was conducted in Aspen plus®, where the electricity consumptions of all devices were extracted as “electricity flows” and summarised using a “Mixer” module to yield a total flow of “electricity consumed” as output and reported in the excel sheet B. We have now breakdown the total electricity consumed according to unit processes, such as pretreatment, HTC, AD, Boiler, and Combustion, as listed in Excel Sheet B, Columns K to O.